# Itaconic Acid and Its Applications for Textile, Pharma and Agro-Industrial Purposes

Nisha Devi [1], Shubhangi Singh [1], Shivakumar Manickam [2], Natália Cruz-Martins [3,4,5,6,*], Vinod Kumar [7], Rachna Verma [8] and Dinesh Kumar [1,*]

1 School of Bioengineering and Food Technology, Shoolini University of Biotechnology and Management Sciences, Solan 173229, Himachal Pradesh, India
2 Petroleum and Chemical Engineering Department, Faculty of Engineering, University Teknologi Brunei, Bandar Seri Begawan BE1410, Brunei
3 Faculty of Medicine, University of Porto, 4200-319 Porto, Portugal
4 Institute for Research and Innovation in Health (i3S), University of Porto, 4200-135 Porto, Portugal
5 Institute of Research and Advanced, Training in Health Sciences and Technologies (CESPU), Rua Central de Gandra, 1317, 4585-116 Gandra, PRD, Portugal
6 TOXRUN—Toxicology Research Unit, University Institute of Health Sciences, CESPU, 4585-116 Gandra, CRL, Portugal
7 School of Water, Energy and Environment, Cranfield University, Cranfield MK43 0KL, UK
8 School of Biological and Environmental Sciences, Shoolini University of Biotechnology and Management Sciences, Solan 173229, Himachal Pradesh, India
* Correspondence: ncmartins@med.up.pt (N.C.-M.); dineshkumar@shooliniuniversity.com (D.K.)

**Abstract:** Itaconic acid (IA) is a well-known bio-based monounsaturated organic acid ($C_5H_6O_4$), with a white color and crystalline structure. It is widely used in the agro-based, plastics, textile, paint and pharmaceutical sectors, owing to its flexible structure, due to the presence of functional groups with covalent double bonds. IA is an alternative to the petrochemicals acrylic and methacrylic acids. Commercial manufacturing of IA using *Aspergillus terreus* is more economically effective and feasible, and the Department of Energy (DOE) of the United States added IA under the "top 12" organic chemicals in 2004. This review provides an overview on the synthesis of IA and improvement of its yield by mutagenesis and metabolic engineering of *Aspergillus* and other fungal strains, along with its wide applications for food, pharmaceutical and textile purposes.

**Keywords:** itaconic acid; metabolic engineering; *Aspergillus terreus*; market scenario

## 1. Introduction

With a renewed interest in sustainable development, the chemical industry is making many efforts to replace petrochemical-based monomers with natural ones. Itaconic acid (2-methylidenebutanedioic acid) (IA) is one of the more valuable organic acids and is an important platform of chemical compounds. IA is a crystalline, white color, unsaturated dicarboxylic acid with a methylene group connected to one carboxyl group generated by microorganisms and has industrial importance as a precursor of polymers and chemicals [1]. Citraconic and mesaconic acids are isomeric with IA and can be substituted with acrylic or methacrylic acid. At moderate temperatures, IA can sustain acidic, neutral and slightly basic conditions [2,3]. The US Department of Energy in 2004 considered IA an added-value product from biomass, utilized as a precursor of polymers and chemical intermediates. These include synthetic fibers, resins, paints, acrylic plastics, acrylate latexes, super-absorbent materials, anti-scaling agents, styrene, 2-methyl-1,4-butanediol and 3-methyltetrahydrofuran [4].

IA has a very large market potential [5]. Currently, the industries use pure glucose or sucrose to produce IA at a very high cost; hence, it cannot compete and is considered for potential use in food applications [6]. Nonetheless, it is necessary to use low-cost

agricultural waste as a raw material to regulate and reduce the substrate cost for its use for food purposes [7]. The cost of primary raw materials, competing among the food and biorefinery industries, environmental security, and the production of upgraded products from biowaste benefit from converting these biowastes into IA [8]. Starchy biomass from sweet potato, cassava, sago, corn and sorghum has been reported to generate IA by fermentation [9,10]. IA is a microbially produced organic acid that may be used as a substitute for many petroleum-based compounds, including acrylic and methacrylic acids, and, thus, can help to promote environmental sustainability. IA is renewed rapidly, unlike petroleum, which can lessen the reliance on petroleum and its negative environmental consequences [11,12]. The US-DOE 2004 produced a list of the "top 12" building organic chemicals to promote the bio-based economy, and IA is one of them [13]. Because IA has a vinyl group, it may be used to build poly-(IA), a polymer of IA, which has a wide range of industrial applications [14].

In this sense, this review intends to provide an overview on the manufacturing of IA from agricultural and horticultural wastes produced from various food industries, as well as the current state of research on IA production, covering the selection of substrate, fermentation with microbial culture, down-streaming, process intensification and the future prospects.

## 2. Chemical Route for IA Production

As described in Figure 1, IA was discovered in 1837 by Baup as "Citric acid" (Pfizer & Co.) [15,16]. Crassus (1840) named the same substance that results from the heat breakdown of citric acid as "IA" [16]. IA as an ethyl ester that was polymerized for the first time in 1873 by Swart [15]. Kinoshita [17] was the first to report on the microbial synthesis of IA. Under surface culture conditions, a thread-like colony of fungus was isolated from the salted plum juice, comprising concentrated sugar solutions and chlorides in high concentrations producing an amount of IA up to 0.24 g IA/g substrate, using *Aspergillus itaconicus* for its manufacture [17,18].

Thermal degradation of citric acid and hydrolysis of the anhydrides were the earliest methods used for IA production [19]. Another method used was the decarboxylation of aconitic acid [20,21], and the resultant product "IA" was named as an anagram of the source. The carbonization of citric acid and successive treatment of its anhydride with water are the most common methods of chemical synthesis [22] or using water, propargyl chloride, nickel carbonyl and carbon monoxide, according to the Montecatini (Italy) technique [23]. Tate et al. [3] reported that no other chemical methods could compete with fungal cultivation; hence, they are not used commercially. No more than 0.09 g/g of citric acid has been recorded as an IA yield [24], considered very low in industrial production. The primary disadvantage of chemical production of IA is that it depends upon high-cost raw materials for manufacture and on high temperatures during the reaction, making the process more costly [12].

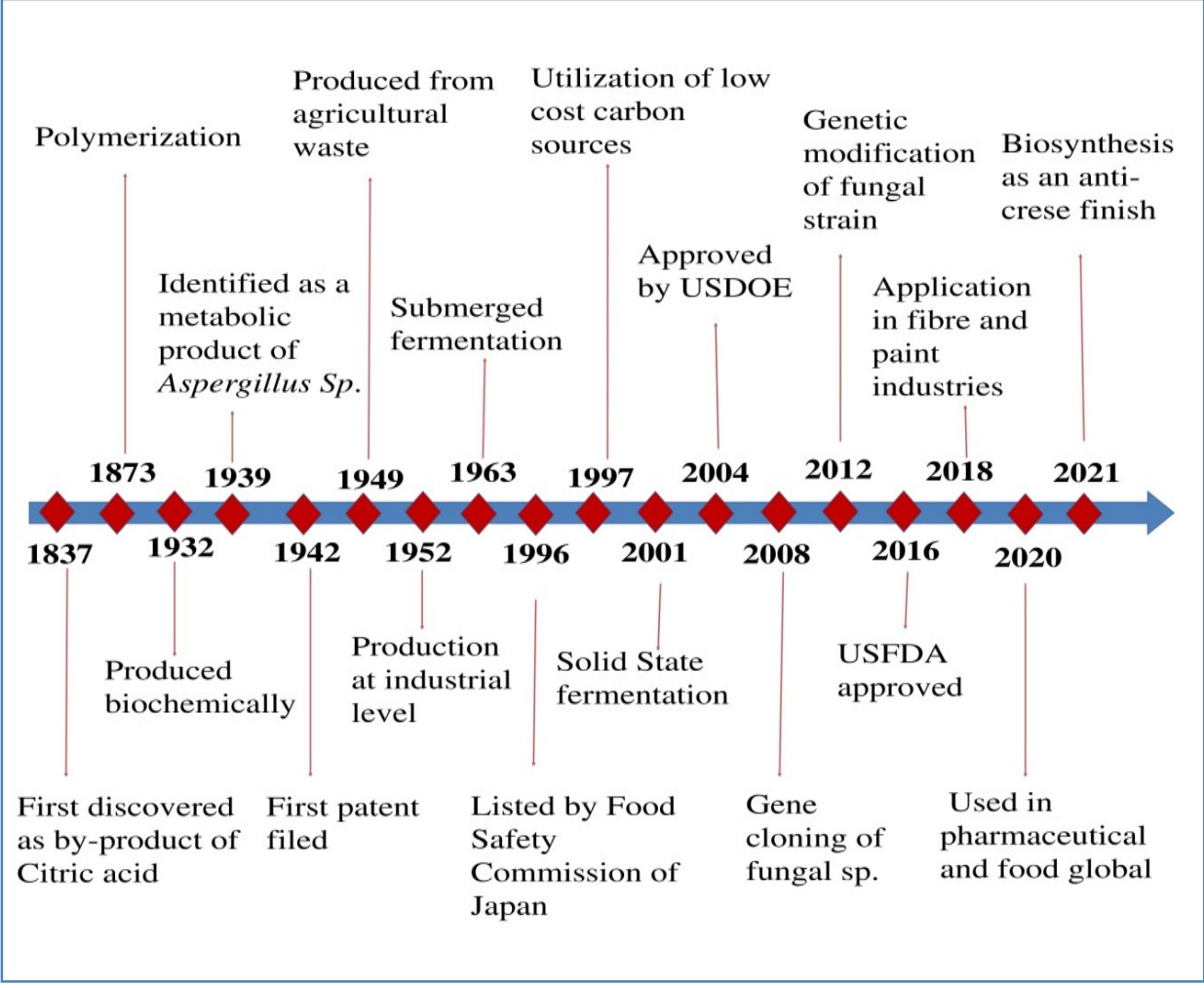

**Figure 1.** Timeline graph for the developmental milestones of itaconic acid discovery and its modification related to production (1837–2021) [15–18].

## 3. Biological Route of IA Synthesis

The biotechnological production of IA using *A. itaconicus* was first described by Kinoshita [17], despite that when using *Aspergillus terreus* a higher final IA content is achieved when compared to other fungal species [25]. In 1945, Charles Pfizer filed the first patent for IA manufacturing on an industrialized level, and, after 10 years, the first production plan was started in the USA. The production of IA using *A. terreus* has been repeatedly improved to be a highly stable practice in which the chemical process could not compete with the biological process for a long time [26].

Nonetheless, the research for the synthesis of IA through enzyme participation is currently receiving greater attention. In the microbial biosynthesis pathway, several sugars, such as xylose, arabinose and glucose have been used as a source of carbon substrates. As revealed in the pattern of *A. terreus*, these sugars are transformed into pyruvate through hexose mono-phosphate shunt and glycolysis [27]. The pyruvate is transferred to the mitochondria from the cytosol and converted into *cis*-aconitate and citrate, as intermediate compounds of the Krebs's cycle. Then, with the assistance of cadA, the *cis*-aconitate is converted into itaconate when the mitochondrial tricarboxylate transporter (Mtt) transports *cis*-aconitate back to the cytosol [28] (Figure 2). To date, IA is commonly produced at an industrial level using the filamentous fungus, *A. terreus*, as it generates the highest output as well as increases the final amount of IA under optimum growth conditions in the fermentation broth compared to other fungal strains [29]. It is noteworthy that,

despite *A. terreus* being the main microbial species used for IA production, according to the requirements of *A. terreus* for optimum growth, it is not the most effective. Because a continuous supply of oxygen is necessary, this results in high NADH levels. A high level of NADH inhibits the major enzymes that play a significant role in IA production. The vigorous stirring during fermentation can readily damage the mycelia grown on *A. terreus* [28]. Moreover, due to the limited activity of natural *cis*-aconitate decarboxylase, CadA from *A. terreus* is over-expressed in these modified strains, but the final yields still remain less [30].

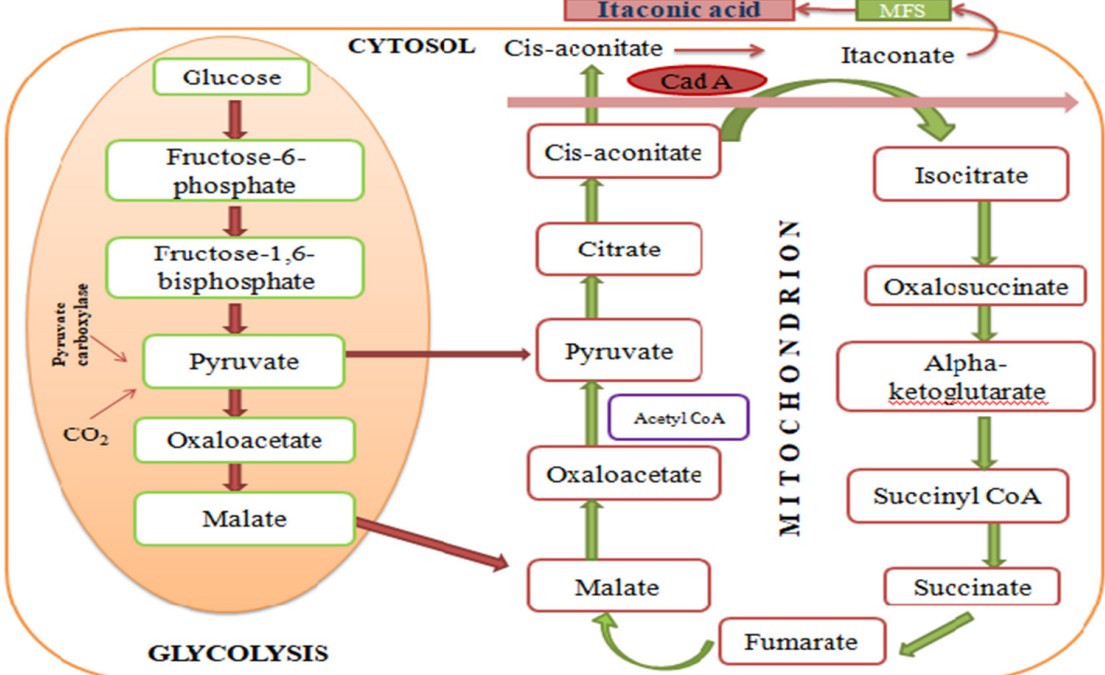

**Figure 2.** Biological route for IA synthesis.

## 4. Genetic Modifications of *Aspergillus* Strain for the Production of IA

### 4.1. Metabolic Engineering

In this biosynthesis pathway, *cis*-aconitic acid decarboxylation into IA is the unique and critical process. Kanamasa et al. [31] observed the characterization of the cadA gene in this biosynthesis; genetic modification of *A. terreus* into another microbe became feasible. Researchers tried to produce IA with *A. niger*, which was selected owing to its great ability to synthesize citric acid (CA) in large amounts of about 200 g/L, and the route for the synthesis of CA and IA are intertwined [32]. However, due to the lack of CadA, the synthesis of IA from *A. niger* does not occur spontaneously; as a result, CadA has been taken from *A. terreus* and transported into *A. niger*. IA yield from the modified *A. niger* strain was initially low [27]. For *A. niger*, codon optimized the cadA gene, and the Mtt and the major facilitator (Mfsa) genes were added, resulting in a 20-fold increase in the final yield of IA. Figure 3 shows that Mtta and Mfsa are important components in synthesizing IA [28]. In *A. niger* strain AB 1.13, the cadA gene from *A. terreus* is expressed [33]. The *A. niger* gpdA promoter has been used to regulate the cadA gene. For this reason, it allows a constitutive expression. A strain of A. *niger* that expresses the single cadA gene may generate around 0.7 g/L of IA, a yield that is not comparable to existing *A. terreus* production strains, but it is a promising starting point for further modification steps. Furthermore, researchers have attempted to enhance the number of expressed genes (Table 1), such as those already identified in the cadA gene and mitochondrial carrier protein (MCP) [25,33].

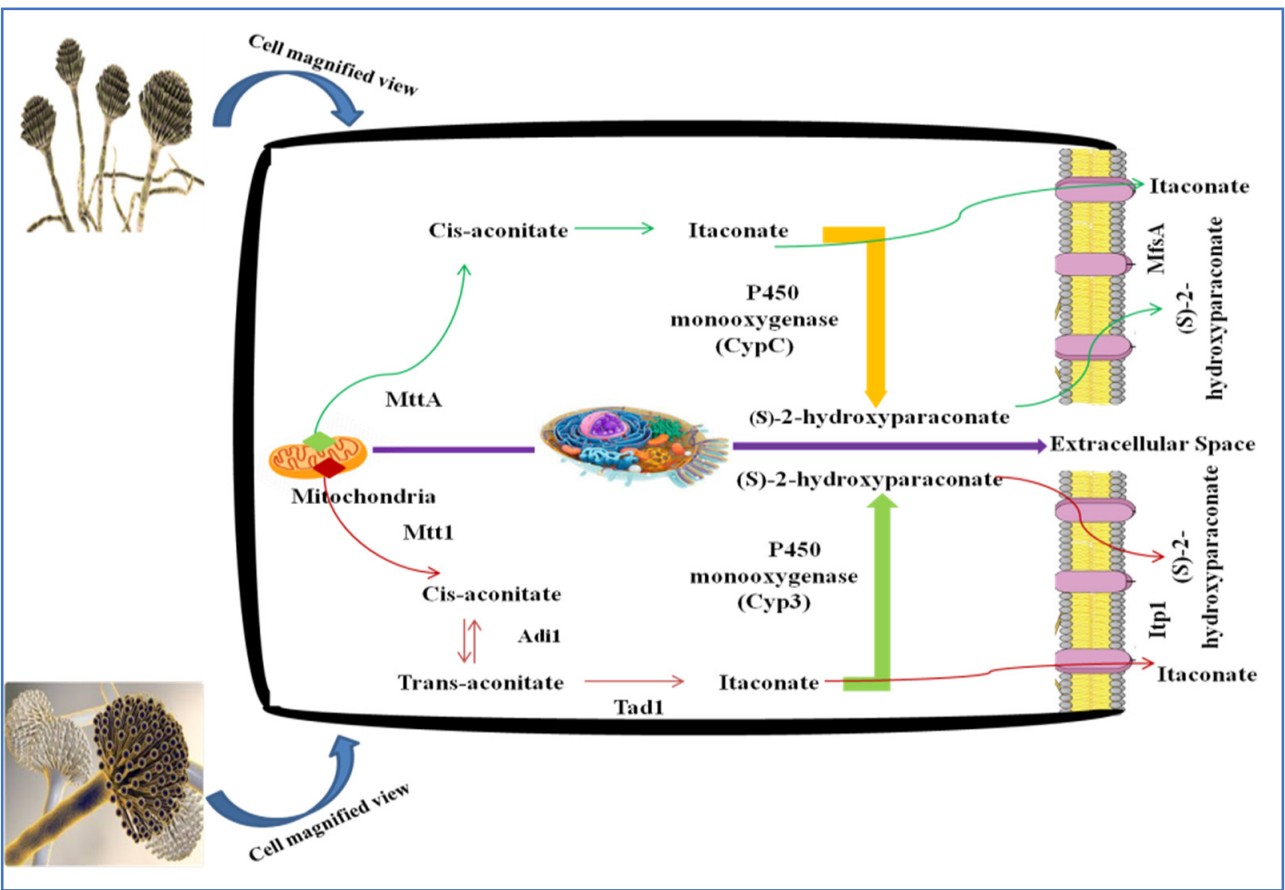

**Figure 3.** Genetic modifications for IA production using *A. niger*.

The yield of IA generated by *A. terreus* is approximately 85 g/L. However, this amount cannot be compared to the synthesis of CA, which can easily attain a concentration of more than 200 g/L in industrial production. Li et al. [33] stated that the IA synthesis is performed in order to produce a theoretical yield of 240 g/L, and, thus, breeding the current strains or focused genetic engineering might help achieve this objective. The activity of 6-phosphofructo-1-kinase is inhibited by adenosine triphosphate (ATP) and citrate, a gene that influences the efficiency of IA synthesis in *A. terreus*. Capuder et al. [41] investigated the shortened *A. niger* pfkA gene in order to reduce the citrate inhibition, and a higher CA yield was obtained using ATP. When expressed in *A. terreus*, the shorter pfkA variant positively affected the quantity of IA [42].

Lin et al. [43] showed that the strains produced a better yield after interruption of aeration. However, it was realized that the genetic make-up of *A. terreus* is insufficient to support the production of higher amounts of IA. So, a strategy of IA production through genetic engineering into another host organism already familiar enough to support the high amount of IA production was suggested, and *A. niger* was best suited [28].

**Table 1.** Role of genetically modified non-*Aspergillus* genera in the production of IA.

| Fungal Strain | Gene Used | Enzymes/Protein | Modification Pathway | References |
|---|---|---|---|---|
| *Candida lignohabitans* CBS 10342 | cadA | cis-Aconitate decarboxylase | Heterologous expression of cadA under the control of GAP promoter and terminator | [34] |
| *Pseudomonas putida* | tad1, adi1 | trans-Aconitate decarboxylase, Aconitase-Δ-isomerase | Heterologous expression of tad1 and adi1 from U. maydis; deletion of PHA synthetases phaC1 and phaC2 | [35] |
| *A. niger* AB 1.13 | mttA, mtt1 | Mitochondrial tricarboxylate transport protein | Over-expression of acl12 (ATP-citrate lyase), citB (Cytosolic citrate synthase), cadA, mttA and mfsA | [36] |
| *U. maydis* MB215 | cyp3,cypC, ria1 | P450- monooxygenase (Regulatory gene of itaconic acid gene cluster) | Deletion of byproduct genes (cyp3); over-expression of ria1, Over-expression of native/regulator rai1 and mttA under Petef promoter (from *A. terreus*); deletion of cyp3 and fuz7 | [37,38] |
| *Pichia stipitis* FPLUC7 | acoA, acnB | Aconitase | Heterologous expression of CAD and over-expression of native truncated ACO (without mitochondrial signal) | [39] |
| *Pichia kudriavzevii* YB4010 | Icd | Isocitrate dehydrogenase | Heterologous expression of At_cad, over-expression of native Pk_mttA and deletion of icd (isocitrate dehydrogenase) | [40] |

*4.2. Mutagenesis of Aspergillus Strain*

Genetic alteration and mutagenesis have boosted IA production in *A. terreus* strains. After repeatedly altering the wild-type *A. terreus* strain IFO6365, Yahiro et al. [44] identified the strain known as *A. terreus* TN484-M1, which, after 6 days of fermentation, generated up to 82 g/L IA. Using the same strain, Dwiarti et al. [9] investigated the IA production from sago starch and obtained about 48.2 g/L of IA. The enhanced performance was thought to be caused by the mutant strain's transcription of CAD1 (the gene encoding cis-aconitic acid decarboxylase), which was five times stronger than IFO6365 [31].

Reddy and Singh [45] developed two mutants of *A. terreus* SKR10 by applying different chemicals or UV mutagens alone or in combination, which are *A. terreus* N45 and *A. terreus* UNCS1, producing 50 and 32 g/L of IA from corn starch and fruit waste extracts, respectively, as compared to the parent strain, which produced 31 and 20 g/L. A modified pfkA gene from *A. niger* was inserted into the genome of a wild-type strain of *A. terreus* [42]. This modified pfkA gene was encoded for a shorter and more active 6-phospho-fructo-1-kinase enzyme fragment, and the obtained transformants accumulated higher amounts of IA than the native strain, albeit after a longer lag phase. As a result, the IA yield obtained from the best transformant A729 was 45.5 g/L, which was more than twice that achieved by the parent strain (21.35 g/L) [42].

**5. Fermentative Production of IA**

Various microbes have been tested and examined for their ability to produce IA. The parameters for producing IA by submerged fermentation (SF) by *A. terreus* are identical to those for producing CA by *A. niger*. These factors include an oversupply of rapidly fermentable carbon sources and high quantities of dissolved oxygen [30]. El-Imam and Chenyu [11] observed that a native strain *A. terreus* NRRL 1960 is the most extensively used strain in IA generation. It is also known as QM6856, DSM826, IFO6123, WB 1960, CBS116.46, IMI44243 and IAM2054 in different culture collections. According to Kuenz et al. [30], this strain may generate up to 91 g IA/L from glucose. Another local strain, *Aspergillus terreus* IMI282743, produced 5.76 g IA/L from palm oil industry waste [46]. At the same time, Elnaghy and Megalla [47] found about 54 g IA/L in fermentable media containing 15% glucose from Egyptian wild-type strains. Despite these yields, the demand for even greater

production has prompted strain development, and the bulk of high-yielding cultivars available today are mutated strains [46,47].

According to Lockwood [48], IA is commonly produced from *A. terreus* grown under phosphate-limited conditions, and a review of patent reports for the production of IA by the *Candida* mutant strain and *Rhodotorula* species, when cultivated under phosphate-limited circumstances [1,49], showed that the *Candida* sp. strain generated up to 35 g IA (35% yield) from glucose [50] and 42 g IA from a *Candida* mutant in 120 h [51]. However, a species of *Rhodotorula* produced only 15 g IA after 168 h [52]. Under nitrogen-limited growth circumstances, William et al. [53] demonstrated that the NRRLY-7808 strain of *Pseudozyma antarctica* generated IA from carbohydrates and various sugars [54]. However, Willke and Vorlop [1] discovered that several species of the genus *Ustilago*, basidiomycetes, generated IA during fermentation. In submerged cultivation, a yeast-like single cell was developed by *Ustilago maydis* and was dominant under high osmotic conditions. Furthermore, *U. maydis* may grow on the hemicellulosic content of pretreated beech wood. As a result, combining filamentous fungi with yeasts gains advantages [55], and *U. maydis* acts as an alternative for IA production. Ramesh and Sastry [56] found that, under limited conditions, it has the potential to produce a maximum titer of IA (68.36 g/L) from glucose.

Some other *ustilago* species, i.e., the *Ustilago zeae* strain, were identified by Haskins et al. [57] for the production of IA in a fermentation broth. A subsequent selection of this species revealed a production of around 15 g IA/L. The Iwata Corporation of Japan investigated various *Ustilago* species, including *U. maydis*, revealing that they were capable of generating 53 g IA/L from glucose in just 5 days [58,59]. Since filamentous fungi may pose issues in fermenters, yeasts have also been tested for IA generation.

## 6. Raw Material for IA Synthesis

The selection of a low-cost substrate is critical in IA generation. According to Hegde et al. [60], the raw material, or the feedstock or substrate, must be cheap, available full year and interchangeable, such as lignocellulosic waste and starch and sugarcane molasses. It is also significant that the selected feedstock is well suited to the specified microbe since not all microorganisms can metabolize and generate the desired yield. Monosaccharides and disaccharides may be used as primary biomass to manufacture IA to achieve a high yield. However, the costs of these carbon sources, including the overall production cost (glucose ranges from 0.35–0.60 USD/kg), are high [8]. Due to this reason, the demand for more economical substrates, such as sago, sorghum, sweet potato, wheat, potato and cassava, have also been examined. However, the use of such substrates has some limitations, namely that they all are useful sources of human food and animal feed, which can lead to ethical issues [8].

*Agricultural and Horticultural Waste as a Substrate*

Agricultural biomass produced by agro-industries as agriculture residues, such as carrot processing waste, jatropha seed cake, sweet potato, kiwifruit peel, pineapple waste and fruit juice industries' biowastes, such as apple and orange pomace, are available as alternative substrates that are both cheap and do not interfere with food and feed applications (Table 2). Hegde et al. [60] observed that all agricultural-waste substrates contain complex carbohydrates (cellulose and hemicellulose), which require pre-treatment before further processing.

**Table 2.** Production of IA using various types of biowaste and microbes.

| Substrate | Microorganisms | Fermentation Mode | Modifications to Improve IA Yield | pH | Temp. (°C) | Fermentation Time (Days) | Aeration Rate (vvm) | Yield (g/g) | Reference |
|---|---|---|---|---|---|---|---|---|---|
| Beech wood hydrolysate | *A. terreus* NRRL 1960 | Shake-flask | Heterologous expression of hemoglobin gene from Vitreoscilla | 3.4 | 35 | 3 | N/A | ↑ up to 0.30 | [61] |
| Corn stover hydrolysate | *A. terreus* M69 | STR | Plasma-induced mutagenesis | 3.1 | 40 | 5 | 0.5 | ↑ up to 0.36 | [62] |
| Banana waste extract | *A. terreus* Mutant bN 45 | Shake-flask | UV, chemical and mixed mutagenesis | 3.0 | 34 | 6 | 0.2 | ↑ up to 0.35 | [45] |
| Sugarcane bagasse | *A.niger* MTCC281 | Shake-flask | Mitochondrial expression of *A. terreus* CadA gene | 4.0 | 35 | 5 | N/A | ↑ up to 0.82 | [63] |
| Wheat bran hydrolysate | *A. terreus* CICC40205 | STR | UV-induced mutagenesis | 3.2 | 32 | 3-5 | 0.5 | ↑ up to 0.41 | [64] |
| Wheat chaff hydrolysate | *A. terreus* DSM 23081 | Shake-flask | Mutagenesis of the glucoamylase gene | 3.1 | 33 | 4 | 0. | ↑ up to 0.41 | [65] |
| Glucose | *A. terreus* DSM 23081 | Shake-flask | Optimization of fungal growth conditions: $O_2$ supply, media components to improve the IA productivity | 3.1 | 33 | 7 | 0.2 | ↑ up to 0.62 | [30] |
| Sago starch | *A. terreus* TN 484-M1 | STR | Chemical and enzymatic hydrolysis with optimization of media composition | 2.0 | 40 | 6 | 0.7 | ↑ up to 0.34 | [9] |
| Corn starch | *A. terreus* TN-484 | ALR | To improve the IA production ALR was used compared to STR | 2.0 | 30 | 6 | 2.0 | ↑ up to 0.50 | [66] |
| Corn starch | *A. terreus* NRRL1960 | STR | Enzymatic hydrolysis of corn starch @85DE | 3.4 | 35 | | 1.0 | ↑ up to 0.38 | [67] |
| Glycerol | *E. coli* scvCadA_No8 | Fed-batch | Over-expression and screening of CadA gene into scv gene of *E. coli* | 6.2 | 30 | 3 | 2.0 | ↑ up to 0.98 | [68] |
| Glucose | *Ustilago maydis* MB215 | Fed-batch | Metabolic engineering with a wild strain of *U. maydis* by deletion of *cyp3* and over-expression of regulator gene *ria1* under $P_{etef}$ promoter | 6.8 | 30 | 5 | 0.4 | ↑ up to 0.24 | [69] |
| Watermelon | *A. japonicas* | SSF | DNA sequencing by ITS4 with ITS1 | 6.0 | 37 | 8 | N/A | ↑ up to 0.22 | [70] |
| Empty palm oil fruit bunches | *Aspergillus terreus* K17 | ALR | Pretreatment of enzymatic hydrolysate with steam explosion and saccharification | 2.5 | 30 | 3 | 2.0 | ↑ up to 0.39 | [71] |

In addition, agricultural-waste feedstock has properties that need to be overcome; for example, feedstock composition variation and the conversion rate of the product are much lower in the presence of inhibitors and by-products [8]. Furthermore, agro-waste pretreatment is necessary to improve and optimize the extraction of reducing sugars from lignocellulosic biowaste, as well as to lessen unnecessary intermediate products and inhibitors (furfural and hemi-furfural). Using less expensive substrates such as agro-waste might significantly reduce IA manufacturing costs [8,9]. Table 2 shows the examples of IA manufacturing from agricultural waste streams, such as sugarcane molasses and maize starch.

On the other hand, some potentially hazardous components in the feedstock that can naturally exist or be present as a result of pretreatment methods might influence the fermentation organism's development and production, decreasing yields even further, and toxins in the final product boost costs by demanding costly purifying procedures [20,65]. In the biosynthesis of IA, CA has also been used as a parent component. As CA is less expensive than IA, it may be cost-effective. Although it is widely dispersed, clean, ecological and renewable, lignocellulose is another possible raw material for IA fermentation [72].

Recent studies have examined agricultural biomass/residues such maize cob, rice bran, wheat husk and straw for the production of IA. IA cultivation using sorghum bran acid hydrolysate showed that cellulase had no influence on *A. terreus* growth, although the yield of IA was only one-fourth of that of using glucose [10]. The effectiveness of the NRRL1960 strain of *A. terreus* to generate xylanase and IA from the cotton stalk, maize cob and sunflower stalk was evaluated by Kocabas et al. [73].

Two strains of *A. terreus* were employed to develop the IA in a ten-time dilution of corn cob and wheat bran crude extract with the addition of glucose, but no IA was produced [74]. Pedroso et al. [75] examined that 1.9 g IA was produced from partially purified phosphoric acid crude rice bran extract at 49 mg/g sugar. Furthermore, Krull et al. [65] obtained 0.6 g/L IA extracted from wheat husk crude extract and treated with sodium hydroxide at room temperature before washing. The detoxified hydrolysate yielded 27.7 g/L IA. In addition, other researchers looked into IA fermentation using forest waste as a substrate [65,75].

Klement et al. [55] revealed that the maximum amount of cellulose and hemicellulose from beech wood might be used for IA synthesis. In addition, Saha [27] discovered that *A. terreus* could synthesize IA from xylose and arabinose, thereby completely using hemicellulose and cellulose in cellulosic biomass. Tippkotter et al. [61] discovered that in the enzymatic hydrolysis of beech wood prepared with an organic solvent system, *A. terreus* NRRL 1960 could not reproduce without detoxification. Yang et al. [76] studied possible IA production from bamboo wastes. They discovered that bamboo hydrolysate that had been pretreated and enzymatically hydrolyzed might not be directly used for IA synthesis and growth of *A. terreus* [27,55,61,76].

## 7. Factors Affecting IA Production

IA production in *Aspergillus terreus* is highly dependent on many parameters, including oxygen supply, the concentration of phosphate and presence of metal ions, pH and flow rate [30]. This implies that product yield will be limited if these parameters are not properly controlled during fermentation.

### 7.1. Effect of Oxygen

Oxygen supply is an important factor for synthesizing IA. The formation of IA with *A. terreus* and a small break of the oxygen supply results in reduced yield or even the production of itaconic acid [30,77–80]. The critical period can be as short as 3 min until a negative effect occurs. It might vary depending on how the fermentation settings are used. Kuenz et al. [30] showed that an adequate and constant oxygen supply is required during the generation phase, regardless of using an IA-producing strain. After an oxygen deficiency, the fungus can restore its ability to generate IA. During the synthesis phase, with *A. terreus*, the concentration of dissolved oxygen affected the production of IA to some

extent and it was observed that the variation in the dissolved oxygen level between 20 and 60% did not impact the concentrations of IA, while a slightly better production was stated at a low level of dissolved oxygen [81]. However, due to the significant shaking frequency followed in many investigations, it may be assumed that a high concentration of oxygen supply is required for the development of *U. maydis* [12,82–84].

### 7.2. Effect of pH

A broad pH range of 2 to 5.9 was used for the initial synthesis of IA with *A. terreus* [44,77,81,85]. A pH ≥ 3 influences conidia germination, forming IA [86,87]. With spores' germination, the ammonium ions are consumed, acting as the nitrogen source and releasing protons, causing a decrease in the pH in the unbuffered media [88]. Kuenz et al. [30] initially adjusted the pH to 3.1 during the production, with the intention that the pH would drop below 2.0, and did not change the pH. An increase in the pH due to a single pH shift/control increased the absolute concentration of IA from 87 to 146 g/L [87]. The maximum production of IA of 160 g/L was achieved under a shift to pH 3.4 in the production phase [65]. The pH dependency of IA was described using *Ustilaginaceae* [89].

### 7.3. Effect of Temperature

Temperature is a key physical component influencing the development of fungal species. Meena et al. [90] discovered that the external temperature significantly impacts cell growth, metabolism and the synthesis of IA. *Aspergillus* species were first cultivated in the temperature range of 15 to 45 °C. At 35 °C, *A. terreus* produced the highest amount of IA (26.10 g/L), followed by *A. niger*, *A. flavus* and *A. nidulans*, which produced 23.5, 17.90 and 16.90 g/L, respectively. The highest production of IA was reported up to 35 °C, with a slight reduction observed when the temperature increased to 38 °C.

### 7.4. Effect of Incubation Time

Increased incubation time leads to decreased IA synthesis. It could be related to nutrient scarcity and the storage of pollutants. The growth of fungal strains is often inhibited by an accumulation of organic acids in the fermentation media. Rafi et al. [91] examined the factors that affect the incubation period of IA production and noted that 120 h is the optimum incubation time for *Ustilago maydis*. In contrast, when using *A. terreus*, a total of 160 h was necessary for obtaining the maximal output of IA. In comparison, the similar *A. terreus* strain took approximately 9 days and 16 h to grow when the reactor volume was increased ten times [65]. As a result, the incubation time to produce IA is proportional to the amount of nutrients added to the fermentation process media; based on this, the maximal incubation time for developing IA using *A. niveus* was 168 h [90].

### 7.5. Effect of Metal Ions

*A. terreus* presents a unique morphological appearance determined by several factors of the medium frequently employed for the commercial synthesis of IA. Zinc, iron, manganese, calcium, cobalt and nickel ions in the medium can significantly impact the morphology of *A. terreus* [30,86,92–95]. It was proposed that high manganese concentrations changed the morphology of the mycelia into being floppy and heavily branched, causing a reduction in IA productivity and quality [93]. Fe(II) has been shown to have a comparable inhibitory effect at doses of more than 2 mg/L. IA production was unaffected by the Zn(II) concentration [96]. As a result, manganese and other heavy metal ions have a very similar effect on the morphology and synthesis of IA in *A. terreus*, similar to *A. niger* [96–98]. Heavy metals disrupt the function of vital enzymes in IA production, such as the essential enzyme CAD and structure [99].

## 8. Downstream Processing of IA Production

Before decolorization and drying, IA must be isolated from the fermentation culture, often through filtration and crystallization. Most IA is recovered during crystallization,

with a typical yield of about 80%. Instead of crystallization, liquid–liquid extraction, precipitation, adsorption and membrane separation could be performed.

### 8.1. Biomass Removal

After fermentation, substrates are converted to IA, and the resultant broth comprises biomass, other organic acids, their minor components and fermentation medium residues. The biomass removal is usually the first downstream stage, followed by the concentration and purification of the acid to the desired level. The acid's initial concentration (recovery) from the broth is essential for IA synthesis. Reducing throughput and high final concentrations minimize overall operating and equipment costs [13]. The purification procedures used for IA are shown in Table 3.

### 8.2. Product Isolation and Purification

#### 8.2.1. Crystallization and Precipitation

Crystallization is a standard method used to recover IA produced by fermentation. Dwiarti et al. [9] observed that, using crystallization, IA was produced from two fermentation broths containing glucose and hydrolyzed sago starch. After purification, glucose and sago starch hydrolysate exhibited a purity of 99.0 and 97.2%, respectively, with 51% of total yields. Okabe et al. [100] claimed an overall process yield of 80% for industrial IA synthesis through crystallization. It is fast and effective but requires expensive and high-energy equipment. However, it gives a pure product [100,101].

IA can also be recovered by precipitation with calcium and lead salts [101]. Another IA precipitation method uses calcium hydroxide [102]. Calcium itaconate formed in this method is less soluble than the acid it precipitates; the solid is recovered by filtration. The IA can be regenerated by reacting calcium itaconate with sulfuric acid and further purified using activated carbon and a subsequent crystallization step. However, a significant quantity of waste calcium sulfate sludge is generated by this recovery technique. Precipitation only works when the cost of the chemicals is low enough or when it is feasible to recycle them, as in the case of CaSO4, which requires expensive calcination to CaO and SO3 [103].

#### 8.2.2. Solvent Extraction

The use of esters, long-chain alcohols and alkanes as solvents for purification of IA is ineffective due to the acid's poor distribution coefficient, i.e., IA has limited solubility in organic solvents compared to water [104,105]. Therefore, organophosphorus compounds or tertiary or quaternary amines, which could alter the distribution coefficient, may be used to purify IA. This method is known as reactive extraction (RE). The acid–extractant combination designed has a strong affinity for the organic phase (solvents known as diluents). The acid in this compound may be recovered, and the extractant can be recycled and used in a subsequent extraction (back-extraction). RE has been investigated and demonstrated to be more effective in recovering organic acids when used with certain extractants and diluents [105]. Due to thermal stability and ease of recovery by simple distillation, aliphatic amines and organophosphates have been investigated as solvents for removing IA from the aqueous phase [103,104,106–109].

#### 8.2.3. Electrodialysis

Electrodialysis (ED) permits a concurrent removal and absorption of ions in solution and is an intriguing option for IA purification [110]. When an electric field is supplied to the anion exchanger membranes, itaconate ions are exchanged, separating IA from other residues in the fermented broth. The resultant dialysate may be recycled to use the non-fermented sugars. ED does not require heating or harmful chemicals and operates in a low-temperature environment [103]. ED is undertaken using monad elements (univalent) electrolytes, transforming IA into di-sodium salt and recovering around 98% of IA; however, the high cost of the membrane used, membrane fouling and inability to extract charged nutrients and inorganic ions are the main disadvantages of ED [111].

#### 8.2.4. Membrane Filtration

In situ product recovery (ISPR) entails the integration of fermentation and large-scale reactive extraction and downstream units, which is an appealing technique for achieving a cost-effective and petrochemically competitive biotechnological process for IA production [105]. Carstensen et al. [82] established a new ISPR technique, known as reverse-flow diafiltration (RFD), using a membrane bioreactor (Table 3). The use of RFD for manufacturing IA from *U. maydis* and *A. terreus* fermentation systems, together with pertraction-based IA purification, produced a ten-fold improvement in output [112].

**Table 3.** Techniques for the purification of IA at a commercial scale.

| Applied Technology | Mechanism of Purification | Key Features | References |
|---|---|---|---|
| Crystallization | Cooling and evaporation at low pH. The crude crystals can be purified by solubilizing and treating with activated carbon at 80 °C | A simple and efficient process gives pure product but demands high energy and costly equipment | [28] |
| Precipitation | Precipitation with calcium and lead salts | Easy to perform and feasible when the price of reagent is low or if it can be recycled | [103] |
| Liquid–liquid extraction | By using solvents such as long-chain alcohols, esters and alkanes | The differential coefficient of organic solvents is low as compared to water | [105] |
| Reactive extraction | By using extractants and diluents such as organophosphorus compounds, or tertiary or quaternary amines | High productivity, increased conversion of substrate to product, assists in maintaining pH without the addition of basic solutions and can make the continuous process more energy efficient | [105] |
| Membrane separation | Reverse flow diafiltration. By using permeable membranes and a concentration gradient | Yields a product stream through hydrophilic ultrafiltration where the hollow-fibre membrane is immersed in the bioreactor. Reverse flow direction with a wash solution to prevent loss of performance and maintain a constant volume. Obtained 100% recovery of pure IA. RFD process minimizes the hydro-mechanical stress that causes wear of the membrane and the risk of oxygen limitations | [82] |
| Adsorption | Using alumina, activated carbon, silica and different kinds of synthetic ion exchange resins | It is more selective than extraction and does not involve solvent emission. Low energy consumption compared to crystallization and membrane separation | [113] |
| Electrodialysis | Recovery of IA through ED with univalent electrolytes converting the acid into a disodium salt. Itaconate anions are transported through the anion membrane with a solute recovery yield of 98% | ED can be used in fermentation integrated with simultaneous removal of IA. Electrodialysis with bipolar membranes (EDBM) can produce IA from itaconate salt and may be used to control the pH of the fermenter. | [114] |

A single crystallization step is not enough to recover all the products in a stream, and there must always be a recycling step. Crystallization is used to crystallize previously concentrated solutions as a polishing process, and, in this scenario, the size, cost and energy needs are reduced [115]. Adsorption has a relatively high capital cost for producing a large quantity of chemicals because of the adsorbent cost. However, it compares positively with extraction as it does not involve solvent emissions and is more selective; it also compares constructively in terms of energy consumption with crystallization and membrane processes and can be used in line with fermenters for enhanced substrate conversion by-product sequestration. If the desorption is further developed using temperature-equilibrium shifts or suitable solvents, the requirement for reagents will be less than precipitation [113].

## 9. Applications of Itaconic Acid

Due to the advancement of biotechnological techniques and reduction in cost, IA was identified as a promising commercial compound [116]. IA is most commonly used in the manufacturing of styrene-butadiene rubber (SBR) latexes, synthetic latex, super-absorbent polymers, chelant dispersing agents and methyl methacrylate. Figure 4 shows the wide applications of IA in food, polyester, resin, detergent and paint industries.

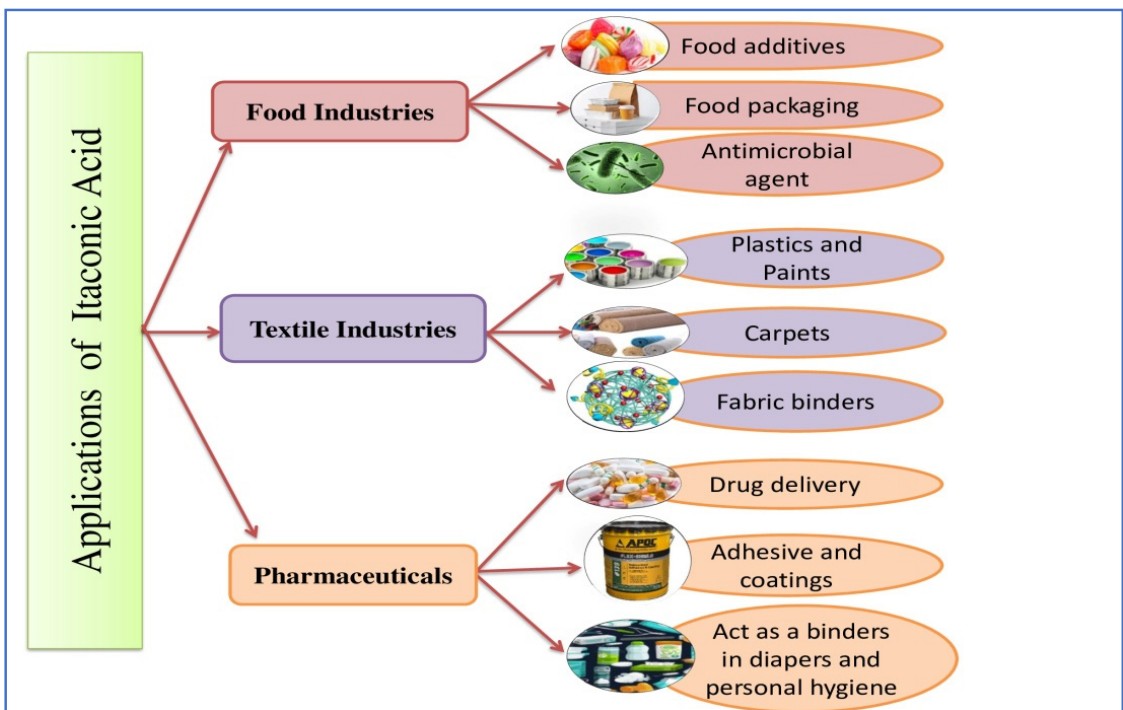

**Figure 4.** Applications of IA and its derivatives in food, textile and pharmaceutical industries.

IA and its derivatives have a unique ability to create hydrogels due to their trifunctional structure. Hydrogels absorb substantial liquid volumes when kept in aqueous solutions due to their cross-linked polymer chain structure [117]. To meet the requirement as a hydrogel, IA has to take up no less than 10% of water [118]. These polymeric networks have the same flexibility as real tissue and do not dissolve in water. Hydrophilic gels exhibit comprehensive liquid behavior, such as a considerable amount of unbound liquid and solids with well-defined features. Soluble molecules can migrate over the gel. Hydrogels may be used in a broad range of products because of their unique properties, including super-absorbent polymers, tissue engineering, contact lenses, drug administration and biotechnology tools [119].

In 1991, Singh et al. [120] published the first instance of polyesters based on itaconic acid by using p-toluenesulfonic acid as a catalyst to create polyesters from itaconic acid and PEG-600 [121]. Hydroquinone was also added as an inhibitor to stop the radical cross-linking of the unsaturated double bonds. These polyesters were building blocks for hydrogel microspheres loaded with biodegradable vaccines. In this instance, the ammonium persulfate aqueous solution catalyzed cross-linking. The rates of vaccine release were then examined on the level of cross-linking [121]. The cyclic derivatives of itaconic acid, itaconic anhydride, and itaconimide homopolymerize by FRP more easily than the acid itself. For starters, because they are charge neutral, they are unaffected by the reaction medium's pH. The alignment of substituents in the ring structure considerably minimizes the steric hindrance of the double bond, which is another factor contributing to their increased reactivity [122].

Yousaf et al. [123] achieved a significant breakthrough in the development and use of bio-based polyesters generated from itaconic acid [123]. Without the use of a catalyst

or inhibitors, they created four distinct polyesters using the thermal polycondensation of itaconic acid, a second dicarboxylic acid, and trimethylolpropane (TMP). This is extremely noteworthy because, in the absence of inhibitors such as hydroquinone or methoxy phenols to squelch the radicals formed at high temperatures, the unsaturated double bond present in itaconic acid often results in undesirable cross-linking. Eventually, as a result of this, the gel begins to develop during the polycondensation process. A low-temperature method for the polycondensation of unsaturated polyesters comprising maleic anhydride and itaconic anhydride was discovered by Takasu et al. [124]. This was accomplished by combining both anhydrides with 3-methyl-1,5-propanediol while including bis(nonafluorobutanesulfonyl)imide (Nf2NH) [121,123].

### 9.1. Textile Industries

Microbially-produced unsaturated polyester resins are being developed to solve these challenges and reduce the amount of vinyl-benzene used. The massive creation of plastic garbage in today's world is a growing source of environmental concern. Growing plastic industries are a rising problem globally, reaching approximately 322 million tons in 2015. Moreover, it is predicted that it may be twice this amount by 2035 [18].

A novel composite material is generated by cross-linking with a reacting solvent, especially dimethyl itaconate, bio-based unsaturated polyester resins (UPR) and chemically modified polyethylene terephthalate (PET) particles. Bio-based UPRs have shown poorer mechanical capabilities but are favorable due to their durability, inexpensiveness and decreased carbon traces [125]. The melt polycondensation of diluted dimethyl itaconate using IA and propylene glycol resulted in 100% green unsaturated polyester resins that can be efficiently synthesized [126]. However, the increase in the amount of itaconate ester groups does not affect the mechanical qualities of the polyester resins. The key feature of tangible bio-based unsaturated polyester resins is that they do not emit impulsive petrochemical and harmful components. As a result, IA and its equivalents have proven to be a low-cost, high-precision tool for producing these polymeric materials [127].

For cotton fabrics, poly-carboxylic acids have been utilized as non-formaldehyde-resistant print finishing reagents. IA is polymerized both in situ and in an aqueous solution on cotton fabric in the presence of a $K_2S_2O_8/NaH_2PO_2$ starting system. The ability to crosslink cotton cellulose with poly (itaconic acid) (PIA) and the polymer created by in situ polymerization of IA provide cotton wrinkle resistance. The effectiveness of cotton fabrics cross-linked by PIA and those cross-linked by in situ polymerization of IA were discovered in a study by Yang et al. [128]. The wrinkle recovery angles of the fabrics after PIA and IA treatments were comparable. However, due to cellulose breakdown, the cotton fabric treated with IA lost more tensile strength than that treated with PIA [128].

### 9.2. Food Industries

The packaging industry aims to protect food from oxygen, microbes, light and other environmental factors. However, the packaging business is currently dominated by petroleum-based synthetic polymers. Active packaging contributes to the expansion of food preservation and the improvement of food quality. The most common use of bio-based IA in the food industry is in active/intelligent packaging, such as smart nano-hydrogels and the discharge of natural food preservatives (Table 4) [18,127,129–132]. Lactic acid derived from plants and its derivatives are also used to make synthetic polymers. Lactic acid, butanediol and IA are transformed into a copolymer, poly (d,l-lactic acid-1,4-butanediol-itaconic acid), using substantial polymerization, and the resultant coating, when applied to the back of a commercially folded boxboard, was entirely grease-resistant and exceeded conventional poly-(lactic acid) coatings. Poly (d,l-lactic acid-1,4-butanediol-itaconic acid) (PLABDIA) is a PLA-based polymer that can be used in various applications, including dry and greasy produce packaging [133]. The environmental viability of using biomass waste as a substitute in manufacturing natural polymers has been raised.

**Table 4.** Properties and applications of IA polymers and their derivatives.

| Material Used | Product | Substitute | Properties of End-Product | Application | Reference |
|---|---|---|---|---|---|
| Sulfonated poly IA | Imidazoline derivatives | Sodium tri-polyphosphate | Produce super-absorbent polymer (SAP) | Detergents and Industrial cleaners | [1] |
| Copolymer of IA | Rubber-like resin | Maleic anhydride | High electronic conductivity | Electrical insulation | [127] |
| N-vinyl-2-pyrrolidone | IA hydrogels | Sodium tri-polyphosphate | Hydrogel formulation with deep eutectic solvents (DES) | Antifungal drug | [134] |
| Styrene-butadiene lattices | Methyl methacrylate | Styrene | Superior adhesion | Carpet backings and paper coatings | [135] |
| Vinylidene chloride | Methyl methacrylate | Maleic anhydride | Changes from elastic to rigid | Paper and cellophane | [103] |
| Acrylic lattice | Methyl methacrylate | Acrylic acid | Improves binding | Nonwoven fabric binder | [135] |
| Poly(ethylene terephthalate) | Coated PET with IA | PET | Improved heat seal strength by about 3.5 times | Food packaging and photography | [1] |
| Acrylic acid homo- or copolymer | Glass ionomer dental cement | Sodium tripolyphosphate | Improve reaction kinetics of polymerisation | Anti-fluorogenic properties, i.e., removing fluoride from tooth enamel | [136] |
| poly(N-isopropylacrylamide) | Subcolloidal nanoparticulated hydrogel | Sodium tri-polyphosphate | Forms super-absorbent microspheres with the high absorption capacity | Drug release during ocular delivery | [137] |
| Polycondensed alkyd resins | Resins | Phthalic anhydride | Improved opacity and reflection | Paint formulations | [138] |

The manufacturing of natural polymers using agricultural waste as a replacement promotes environmental sustainability. The extensive use of rice husk leftovers as an immobilization carrier offers exciting potential uses in the future [139]. Synthesized rice husk has recently been used as a novel, cost-effective polymeric carrier for covalently immobilized enzymes (lipase B from *Candida antarctica*). It has been compared to a standard enzyme carrier in terms of effectiveness (an epoxy methacrylic resin). The cosmetic, polymer and food industries are the most prevalent users of the lipase B enzyme. Lipase B immobilized on rice husk is more efficient than lipase B immobilized in epoxy methacrylic resin in solvent-free polycondensation of dimethyl itaconate and 1,4-butanediol [127,128,139]. The development of molecularly imprinted hydrogels using IA and 2-hydroxyethyl methacrylate exhibits successful biofilm suppression. When IA-based polymers are compared with 2-hydroxyethyl methacrylate-based polymers, it is observed that a significant quantity of *Pseudomonas aeruginosa* biofilm formation was reduced. More research into functional monomer concentrations and optimizations is required before IA can be used on food processing equipment [128,140].

*9.3. Pharmaceutical Industries*

IA and its derivatives are innovative co-monomers that have been used in various studies under a wide range of pH-sensitive microgels in anti-tumor medication release [141–143]. IA is a very hydrophilic antibacterial agent and is competent in building hydrogen bonds with similar groups. Including varying amounts of IA and cross-linking agents can successfully modulate the release of the drug and drug packing volumes. This co-monomer can also release the drug in the gastrointestinal tract due to pH responsiveness [144]. The pH-sensitive and complex nature of the hydrogels can be improved by adding IA to the polymeric chain. At the right pH and media, two carboxyl groups in the IA structure interact with electrostatic repulsion due to their enhanced swelling performance [145]. Hydrogels made of polymers synthesized by IA polymerization inhibit the fungus *Candida albicans*, showing a potential application in vaginal infections, yet not including the use of any antibiotic. The high swelling capacity of IA hydrogel, formed at

pH 7.4–10 due to the carboxylate anion structure, allows for biomedical therapy (neutral pH body fluid) and may also be used to prevent wound infections [144–146].

Under normal conditions, IA-based hydrogels are harmless and 88% recyclable (when buried in soil for 90 days) [127]. Because of an ester bond in the IA hydrogel matrix, biodegradation occurs in hydrogels with an increase in IA content. Under aerobic environmental conditions, various microorganisms, such as fungus, bacteria and protozoa, act on the hydrogels. After degradation, water, methane, $CO_2$ and other compounds are produced [146]. For example, the medicinal effectiveness of ampicillin (antibiotic) encapsulated in nano-hydrogel was proved in a trial on IA-implanted tragacanth gum with ampicillin. The rate of release and response against *E. coli* was more evident: the inhibition zone for the standard/control was 15 mm, while the size and shape of the inhibition zones for the considered nano-hydrogels were 19.3 mm, implying that these nano-hydrogels can be handled under specific antibiotics [147]. Research on paracetamol release, a very soluble and porous drug, revealed an appealing result by free radical co-polymerization of IA and 2-propenamide. An enhancement significantly influences the fluid and drug retention capacity of the investigated biopolymers in IA and a decline in cross-linking agents [144].

The development of anticancer treatments is one promising use of IA nano-hydrogels because cancer is one of the most ubiquitous disease concerns globally [127]. Cancer cells are obstinately unique and may quickly adapt to changing environmental conditions, and chemotherapy is the most commonly used therapeutic approach [148]. The use of nano-based composites particles to enhance anticancer treatment might be one answer to this conundrum because cancer cells have antagonistic effects, and uptake of chemotherapeutic drugs is low at the cellular level. The control of betanin (a natural bioactive substance) and doxorubicin (an anticancer medicine) was assisted using a pH-responsive multi-drug nanocarrier. The pH-responsive methoxy poly (ethylene glycol)-poly (2-dimethylamino) ethyl methacrylate-co-itaconic acid PEG-P(DMAEA-co-IAc) nanoparticles provide controlled drug discharge at the pH values of cancer cells since cancer cells have lower pH values than healthy cells (Table 3) [131,132]. With the synchronous circulation of betanin and doxorubicin, the use of this nano-carrier showed enhanced consequences of cell migration, and the cancer cells' death also progressed [149]. Poly (itaconic anhydride-co-3,9-divinyl-2,4,8,10-tetraoxaspiro (5.5) undecane) (P(ITAU), another bioactive chemo-preventer multi-drug carrier system, was enhanced with quercetin (pro-apoptotic action on cancer cell lines) and diclofenac (nonsteroidal anti-inflammatory drug), and was revealed to be extra imprudent to a biological stimulus when combined with hyaluronic acid [127,149,150]. The best results were obtained with 20% of this matrix, the research using in vivo studies of six different types of hydrogels with different concentrations of P(ITAU). Furthermore, the examined smart gel matrix's swelling point was revealed to be dependent on the pH, temperature and environmental stimuli, confirming crucial properties [151].

Researchers have synthesized minocycline-imprinted hydrogels with well-designed monomers methacrylic acid, acrylic acid, IA, 1N isopropyl acrylamide, 1N ethyl acryl amide and hydroxymethyl acrylamide for the treatment of visual disorders, including dry eye, glaucoma and conjunctivitis [152]. The research findings from the functional monomers used for hydrogen bonding with IA were significant and showed their great potential as hydrogels imprinted with minocycline. However, acrylic acid is a more prevalent monomer utilized in the production of contact lenses with IA. Dosimetry techniques must be precise and trustworthy since they help ionizing radiation treatments be verified. A recent research study for use in radiology examined the water-equivalence characteristics of three different polymer gel dosimeters and the Fricke (cheaper and simpler gel dosimeter) technique. Each polymer application in gel dosimeters, such as IA, N-isopropyl acrylamide and polyacrylamide, showed the same linear trend in a dose-response curve, as well as a higher intensity in water-equivalence [153].

The antibiotic-resistant bacteria are an alarming source of various illnesses, and they take significant casualties (50,000 persons) each year in Europe and the United States [154]. Antimicrobial peptides help each organism's immune system respond to infections quickly.

An ongoing investigation into synthetic antimicrobial peptides is now being improved depending on the environment. These polymeric films must be robust and consistently adhere to medical equipment [155]. A promising material against bacterial infections was provided employing irregular functionalized monomers of di-itaconate copolymerise with N,N-dimethyl acrylamide as an artificial performance of these antimicrobial peptides. IA that has been naturally polymerized is a resource that is secure and natural and can prevent bacterial contamination [156]. The above-described polymers are fairly economical (produced by systems with no metal as an initiator); however, additional basic improvement is still under advancement. IA salts utilized in beverages or foodstuffs operate as a metabolic controller of glycolytic pathway with antiobesity, antidiabetic and/or antilipemic properties [127].

## 10. Market Trend and Future Perspectives

The current market prices for IA and other unsaturated monomers illustrate that IA already competes with acrylic acid and can be used as a drop-in monomer for various purposes. In fact, due to market pricing, it cannot be utilized as a replacement for maleic acid or butenedioic acid. Innovative advancement in fermentation must enhance production and efficiency, while filtration and acid refinement are also progressing [113]. As a result, it shows that rationale improvement will constrain the marketplace.

According to market research, with a production capacity of 80,000 MT/year, IA output was 41,400 MT/year globally. IA had an average price of USD 1800–2000 per tonne in 2011, with a market value of USD 74.5 million [130]. Transparency market research in 2015 listed that the market worth of IA was USD 126 million in 2014, and it is expected to be USD 204 million in 2023, while, according to Global industry and analysis, 2016, the market worth of IA was USD 216.6 million in 2020, and expected annual growth rate is 5.5%, as shown in Table 5 [157]. A recent market data forecast observed that the size and share of the IA market were determined to increase from USD 98.4 million in 2022 to USD 110.4 million by 2027.

**Table 5.** Major producers and suppliers of IA with production capacity (tons/years).

| IA Producing Country | Supplier of IA | Capacity (Tons/Year) | Since Year | Reference |
|---|---|---|---|---|
| USA | Pfizer Food Science | | 1945–1989 | [158] |
| | Cargill | 5000–7000 | 1996 | [159] |
| China | Jiangshan Guoguang | 2000–4000 | 1996 | [160] |
| | Qingdao | 1500 | - | [161] |
| Japan | Iwata Chemicals | 3500 | 1970 | [162] |
| France | Rhodia | 1000–5000 | 1995 | [163] |
| India | Alpha Chemika | 2000 | 2010 | [164] |

The unpredictable and variable assumptions are expected for the IA industry, owing to a lack of application development, even though the Asia-Pacific area is expected to dominate the market in the coming years. The market has now shifted to China, but the supply of sugarcane as a raw material remains an issue since it is growing in demand as a bio-fuel and for consumption in the food and beverage sectors (Grand View Research, 2016). In 2011, styrene-butadiene rubbers accounted for 44% of the IA market (SBR). The most important industrial IA-generating microbes are *A. terreus, U. maydis* and mutant *Candida* species strains. There is active search and expansion in the production of IA and its available technologies. Both these conditions are demanding, whereas the first is important for reducing the cost, and the second is essential as a market constraint [165].

According to the consultancy firm's vision, the manufacture of unsaturated polyesters and methyl methacrylate (MMA) is predicted to be the major growth area for IA, with a potential global production capacity of 407,790 MT in 2020 and a market value of USD 567 million [18]. Demand for IA is driven partly by its economic growth; for example, if bio-based MMA is cost-effective, it can replace approximately 9.25% of ethyl alcohol

cyanohydrins with MMA. Because of the rising demand in the automotive industry, it is also important that anhydride of maleic acid can be used to replace unsaturated polyester resins. According to Weastra, IA can replace up to 5% of maleic anhydride by 2020, as shown in Figure 5 [156].

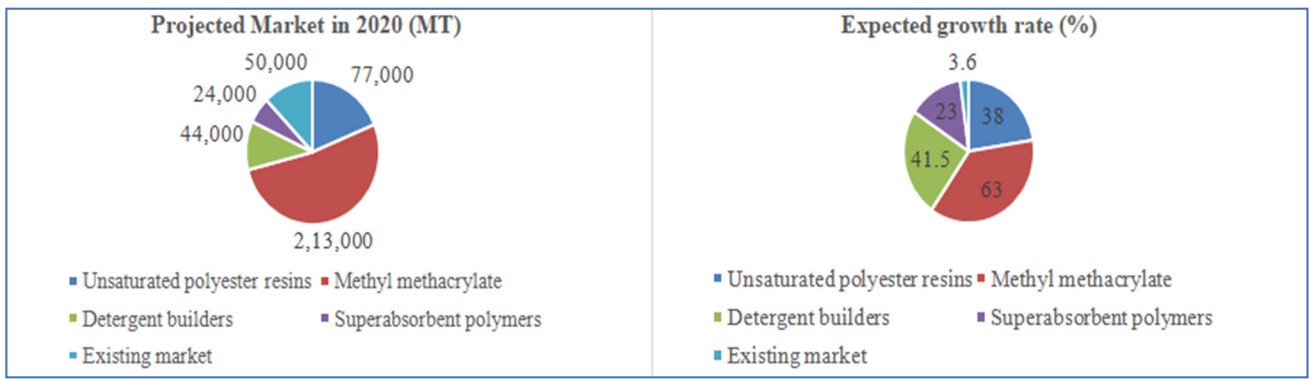

**Figure 5.** IA market projections and growth rate in 2020. Reference: http://www.bioconcept.eu/wp-content/uploads/BioConSepT_Market-potential-for-selected-plateform-chemicals_ppt1.pdf, accessed on 4 February 2020 [164,165].

Efforts for the cost reduction of downstream processing of IA-containing broths are continuing, and conditions for large-scale manufacturing are growing, e.g., the use of air-lift fermenters to assure the high aeration rates required by *Aspergillus* [166]. The synthesis of IA from another resource, such as catalytic disintegration of citric acid, shows more suitability, while the overall concentration produced is lower than IA by fermentation. Between production and application, the conversion of IA into methacrylate is an ongoing research area and, if successful, will primarily increase demand for the bio-based acid, as a result of the competitive methacrylate market [18].

## 11. Conclusions

IA has many industrial uses, but its high manufacturing costs make it uneconomical. IA has broad potential as an alternative for petrochemical-based chemicals, such as acrylic/methyl acrylic acids, and its innovative properties allow for new applications in the polymer, pharmaceutical, textile, food and agro-industrial sectors. As a result, upcoming studies will primarily focus in lowering manufacturing costs by decreasing the raw material costs, leveraging new, affordable and non-food substrates and producing other industrially significant by-products. The biotechnological production of IA needs further research in fungal strain improvement by genetic modification and metabolic engineering. Also, increased IA yield can be achieved through optimization approaches, despite more effective techniques for product recovery being essential to assist in reducing product loss and lower manufacturing costs.

**Author Contributions:** Conceptualization, N.C.-M., V.K. and D.K.; writing—review, N.D.; editing, S.S., S.M. and R.V. All authors have read and agreed to the published version of the manuscript.

**Funding:** This research received no external funding.

**Institutional Review Board Statement:** Not applicable.

**Informed Consent Statement:** Not applicable.

**Data Availability Statement:** No new data were created or analyzed in this study. Data sharing is not applicable to this article.

**Conflicts of Interest:** The authors declare no conflict of interest.

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
