# Peer review of "Itaconic Acid and Its Applications for Textile, Pharma and Agro-Industrial Purposes"

_sustainability, doi:10.3390/su142113777_

Round 1
Reviewer 1 Report
The manuscript content fits well with the journal scopes, and it needs to be revised.
Please correct some grammatical errors which is highlited in the attached pdf.

Author Response
Dear reviewer,
Thanks for your comments, that have markedly contributed to improve the quality of our manuscript. All changes/additions have been highlighted with pink color
Q:1 The manuscript content fits well with the journal scopes, and it needs to be revised.
Answer: Changes as per the reviewer’s suggestion.
Q:2 Please correct some grammatical errors which is highlighted in the attached pdf.
Answer: Changes/corrections as per the reviewer’s suggestion
Reviewer 2 Report
The manuscript presents a review of itaconic acid production methods, and it is strongly focused on biotechnological aspects (which is of course understandable). Chapter 10 deals with the uses of this acid, mainly as a monomer for the production of various types of polymers with applications in textile, pharma and agro-industrial sectors. This part of the review is - in my opinion - the weakest point of the review.
1. Introductory information on the chemistry of itaconic acid polymers - radical and polycondensation - is lacking. Instead, there is a general paragraph on hydrogels (p. 5-6).
There are some review e.g. 10.1039/C6GC00605A, 10.1007/BFb0051282 (and IA derivatives 10.1002/marc.202000546), and papers showing advances in IA chemistry e.g. 10.1016/j.eurpolymj.2017.08.020, 10.1039/D0GC02274H, 10.1002/marc.201900611.
2. Methylene group is not a reason of formation of hydrogels by IA (p. 6). This is an oversimplification.
3. The sentence: “Another promising application of IA is in oncology” (p. 8) is highly imprecise and misleading.
4. Chapter 10.4 is mainly about dimethyl itaconate. Interesting works of C.Q. Yang (e.g. 10.1002/app.12043) on application of IA in cotton processing are missed.
5. Chapter 10.2: “The most common use of bio-based IA in the food industry is in active/intelligent packaging, such as smart nano-hydrogels and the discharge of natural food preservatives [18,125,126, 127, 128].”- however, the cited references 125, 126, 127 and 128 do not concern the acid at all (!).This is MISLEADING the reader.
6. Again, I have comments on a section of the text: “Under normal conditions, IA-based hydrogels are harmless and 88% recyclable (when buried in soil for 90 days). … [137]”. I find it difficult to believe that cross-linked polymers degrade in 90 days in soil, especially as I have NOT FOUND CONFIRMATION of this information in the ref. 137.
Chapter 10 needs to be heavily rebuilt. In this form, it is definitely not a reliable source of knowledge. For this reason my recommendation is major revision.
Author Response
Dear reviewer,
Thanks for the comments raised on our manuscript, they have markedly assisted us to improve the current version of the manuscript. All changes/additions have been highlighted with yellow color.
Reviewer# 2
Q: 1 Introductory information on the chemistry of itaconic acid polymers - radical and polycondensation - is lacking. Instead, there is a general paragraph on hydrogels (p. 5-6).
Answer: Additions have been done as per the reviewers’ suggestion.
Q: 2 Methylene group is not a reason of formation of hydrogels by IA (p. 6). This is an oversimplification.
Answer: Changes as per the reviewer’s suggestion.
Q: 3 The sentence: “Another promising application of IA is in oncology” (p. 8) is highly imprecise and misleading.
Answer: Changes as per the reviewer’s suggestion.
Q: 4 Chapter 10.4 is mainly about dimethyl itaconate. Interesting works of C.Q. Yang (e.g. 10.1002/app.12043) on application of IA in cotton processing are missed.
Answer: Changes/additions have been done in 10.1 as per the reviewer’s suggestion. Chapter 10.1 now becomes chapter 9.1.
Q: 5 Chapter 10.2: “The most common use of bio-based IA in the food industry is in active/intelligent packaging, such as smart nano-hydrogels and the discharge of natural food preservatives [18,125,126, 127, 128].”- However, the cited references 125, 126, 127 and 128 do not concern the acid at all (!).This is MISLEADING the reader.
Answer: Changes/additions have been done as per the reviewer’s suggestion. Chapter 10.2 now becomes chapter 9.2.
Q: 6 Again, I have comments on a section of the text: “Under normal conditions, IA-based hydrogels are harmless and 88% recyclable (when buried in soil for 90 days). … [137]”. I find it difficult to believe that cross-linked polymers degrade in 90 days in soil, especially as I have NOT FOUND CONFIRMATION of this information in the ref. 137.
Answer: Changes/additions have been done as per the reviewer’s suggestion.
Reviewer 3 Report
The manuscript of Saini et al. is a review summarizing the most important achievements concerning the production and utilizations of itaconic acid, an essential platform chemical. I consider that the subject is of scientific interest, since itaconic acid will probably emerge as an essential monomer, replacing petrochemical-based raw materials as intermediate for polymers and various other chemicals. The manuscript is well structured, but recommending it for publication would be possible only following some revisions which I consider necessary.
1. I recommend to merge chapters 4 (Metabolic Engineering used for the production of IA using Aspergillus Strain) and 5 (Mutagenesis of Aspergillus Strain) and to rewrite them in a more expressive way, clearly showing the main outcomes in this topic (utilization of protein engineering tools for improvement of itaconic acid production in Aspergillus strains). Mutagenesis reports for other microbial species, like as Candida, should be included in this chapter, as well. A table would be very useful, to summarize the main strains investigated, techniques used, and productivities obtained. In the present manner everything is a little bit confused.
2. Chapter 9 (Production of IA on an Industrial Scale and its Purification) should be renamed, in my opinion, as “Downstream processing of itaconic acid production” and reorganized, differentiating the main steps of this process (biomass removal, product isolation and purification), with focus on the last one, presenting (in a critical manner) the main techniques described in the literature.
3. Lines 483-487. It is a confusion concerning the utilization of rice husks (not “synthesized rice husks”), since they have nothing to do with “manufacturing of natural polymers using agricultural waste”. In the cited work (reference #130) rice husks are used only as immobilization carriers for the enzyme (a lipase) which catalyzes the synthesis of an itaconate copolymer. The synthesis of natural polymers from biobased itaconic acid and other biobased raw materials, using biocatalysis is obviously a very important research direction, which should be emphasized in this manuscript. In fact, an important review in the topic of itaconic acid-based polyesters, also by biocatalytic way (https://doi.org/10.1039/C6GC00605A, not cited in the manuscript), was published some years ago, but many other articles are available in this subject. I recommend to reconsider chapter 10, including the biocatalytic polymer synthesis from itaconic acid as an important application.
Minor corrections needed:
4. Lines 36-37. Reformulate the sentence “...(IA) is one of the most useful chemical compounds in organic acids”.
5. Lines 47-49. “Currently, the industries use pure glucose or sucrose to produce IA at a very high cost; hence, it cannot compete and is considered for potential food applications”. It is not clear what the authors want to say. Reformulate.
6. Line 97. Replace “biotechnological invention” by “biotechnological production”.
7. Lines 428-430. It is again not clear the meaning of the following sentence “IA is most commonly used in manufacturing 44 % styrene-butadiene rubber (SBR) latexes, 9% synthetic latex, 8% super-absorbent polymers, 7% chelant dispersing agents, and 4% methyl methacrylate”. Moreover, these data must be sustained by a cited source.
Lines 590-591. “IA had an average price of USD 1800-2000 per tonne in 2011...”. This price is more 10 years old and probably has no relevance nowadays. Maybe a more recent price should be useful.
Author Response
Dear Reviewer,
Thanks for the comments raised on our manuscript, they have markedly assisted us to improve the current version of the manuscript. All changes/additions have been highlighted with green color.
Reviewer# 3
Q:1 I recommend to merge chapters 4 (Metabolic Engineering used for the production of IA using Aspergillus Strain) and 5 (Mutagenesis of Aspergillus Strain) and to rewrite them in a more expressive way, clearly showing the main outcomes in this topic (utilization of protein engineering tools for improvement of itaconic acid production in Aspergillus strains). Mutagenesis reports for other microbial species, like as Candida, should be included in this chapter, as well. A table would be very useful, to summarize the main strains investigated, techniques used, and productivities obtained. In the present manner everything is a little bit confused.
Answer: We have created the new heading “Genetic modifications of Aspergillus Strain for the production of IA” under this heading two new sub-sections have been assigned “Metabolic Engineering” and “Mutagenesis of Aspergillus Strain” for more clarity to readers. We have also added a new table as per the reviewer’s suggestion.
Q: 2. Chapter 9 (Production of IA on an Industrial Scale and its Purification) should be renamed, in my opinion, as “Downstream processing of itaconic acid production” and reorganized, differentiating the main steps of this process (biomass removal, product isolation and purification), with focus on the last one, presenting (in a critical manner) the main techniques described in the literature.
Answer: Changes have been done as per the reviewer’s suggestion. Now Chapter 9 becomes chapter 8.
Q: 3. Lines 483-487. It is a confusion concerning the utilization of rice husks (not “synthesized rice husks”), since they have nothing to do with “manufacturing of natural polymers using agricultural waste”. In the cited work (reference #130) rice husks are used only as immobilization carriers for the enzyme (a lipase) which catalyzes the synthesis of an itaconate copolymer. The synthesis of natural polymers from biobased itaconic acid and other biobased raw materials, using biocatalysis is obviously a very important research direction, which should be emphasized in this manuscript. In fact, an important review in the topic of itaconic acid-based polyesters, also by biocatalytic way (https://doi.org/10.1039/C6GC00605A, not cited in the manuscript), was published some years ago, but many other articles are available in this subject. I recommend to reconsider chapter 10, including the biocatalytic polymer synthesis from itaconic acid as an important application.
Answer: Changes/additions have been done as per the reviewer’s suggestion.
Minor corrections needed:
Q:4. Lines 36-37. Reformulate the sentence “...(IA) is one of the most useful chemical compounds in organic acids”.
Answer: Changes as per the reviewer’s suggestion.
Q:5. Lines 47-49. “Currently, the industries use pure glucose or sucrose to produce IA at a very high cost; hence, it cannot compete and is considered for potential food applications”. It is not clear what the authors want to say. Reformulate.
Answer: Changes as per the reviewer’s suggestion.
Q:6. Line 97. Replace “biotechnological invention” by “biotechnological production”.
Answer: Changes as per the reviewer’s suggestion.
Q:7. Lines 428-430. It is again not clear the meaning of the following sentence “IA is most commonly used in manufacturing 44 % styrene-butadiene rubber (SBR) latexes, 9% synthetic latex, 8% super-absorbent polymers, 7% chelant dispersing agents, and 4% methyl methacrylate”. Moreover, these data must be sustained by a cited source.’
Answer: Changes as per the reviewer’s suggestion.
Q:8 Lines 590-591. “IA had an average price of USD 1800-2000 per tonne in 2011...”. This price is more 10 years old and probably has no relevance nowadays. Maybe a more recent price should be useful.
Answer: Changes as per the reviewer’s suggestion.
Round 2
Reviewer 2 Report
My commentsAuthor Response
Dear reviewer,
Thanks for the overall constructive comments
Reviewer 3 Report
I consider that the manuscript was revised in accordance with the observations and suggestions of the reviewers and can be published in the present form. I have only one suggestion, since I consider the statement "Itaconic acid (2-methylidenebutanedioic acid) (IA) is one of the more valuable chemical compounds discovered in organic acids" still not appropriate. I recommend changing it to "Itaconic acid (2-methylidenebutanedioic acid) (IA) is one of the more valuable organic acids and important platform chemical", or something similar (lines 35-37).
Author Response
Dear reviewer,
The sentence was changed according to your suggestions.